# Skin-Resident Memory T Cells: Pathogenesis and Implication for the Treatment of Psoriasis

**DOI:** 10.3390/jcm10173822

**Published:** 2021-08-26

**Authors:** Trung T. Vu, Hanako Koguchi-Yoshioka, Rei Watanabe

**Affiliations:** 1Department of Cutaneous Immunology, Immunology Frontier Research Center, Osaka University, Osaka 565-0871, Japan; trungvu0406@derma.med.osaka-u.ac.jp; 2Department of Dermatology, Course of Integrated Medicine, Graduate School of Medicine/Faculty of Medicine, Osaka University, Osaka 565-0871, Japan; stain_way@yahoo.co.jp; 3Department of Integrative Medicine for Allergic and Immunological Diseases, Course of Integrated Medicine, Graduate School of Medicine/Faculty of Medicine, Osaka University, Osaka 565-0871, Japan

**Keywords:** skin-resident memory T cells, human, psoriasis, cytokines, autoantigens, treatment

## Abstract

Tissue-resident memory T cells (T_RM_) stay in the peripheral tissues for long periods of time, do not recirculate, and provide the first line of adaptive immune response in the residing tissues. Although T_RM_ originate from circulating T cells, T_RM_ are physiologically distinct from circulating T cells with the expression of tissue-residency markers, such as CD69 and CD103, and the characteristic profile of transcription factors. Besides defense against pathogens, the functional skew of skin T_RM_ is indicated in chronic skin inflammatory diseases. In psoriasis, IL-17A-producing CD8^+^ T_RM_ are regarded as one of the pathogenic populations in skin. Although no licensed drugs that directly and specifically inhibit the activity of skin T_RM_ are available to date, psoriatic skin T_RM_ are affected in the current treatments of psoriasis. Targeting skin T_RM_ or using T_RM_ as a potential index for disease severity can be an attractive strategy in psoriasis.

## 1. Introduction

Once the immune system encounters antigens, memory T cells are generated from the naïve T cells and facilitate a prompt response to the re-exposure of the same antigens. Two populations of memory T cells have been defined from human blood circulation: effector memory T cells (T_EM_) and central memory T cells (T_CM_) [1]. T_EM_ are also dominant in peripheral non-lymphoid tissues and T_CM_ have an affinity for secondary lymphoid organs [2,3]. Furthermore, research on murine infectious disease models has revealed that a subpopulation of T_EM_ found in peripheral tissues remain in the same tissues for long periods without recirculation after cure of infection [4,5,6]. These findings led to the establishment of the new population of memory T cells, tissue-resident memory T cells (T_RM_).

T_RM_ are superior to their circulating memory counterparts in their ability to provide the local adaptive cellular defense [7,8,9,10,11]. They can respond to the local antigen re-exposure without the recruitment of circulating T cells to the tissue [12]. In addition, recent studies suggest T_RM_ also contribute to systemic immune responses upon subsequent exposure to specific antigens by proliferating and baring circulating populations, such as T_CM_ and T_EM_ [13,14].

The existence and functional activities of T_RM_ were initially investigated in barrier tissues, such as the gut [6,15], skin [4,5,12,16,17], respiratory tract [18,19], and reproductive tract [20,21], in the context of local defense against pathogens in infectious diseases. However, their roles are now recognized in various conditions, including cancer immunity, tissue-specific autoimmune diseases, and chronic inflammatory diseases both in barrier and non-barrier tissues [22].

Skin T_RM_ are among the intensively studied T_RM_ populations not only in murine models but also in humans. The human skin contains an estimate of 20 billion T cells, doubling those in the circulation [23], and over half of these T cells show the T_RM_ phenotype [24]. Besides infectious diseases, the involvement of skin T_RM_ has been reported in allergic contact hypersensitivity [25]; fixed drug eruption [26]; cutaneous malignancies, including malignant melanoma [27,28] and cutaneous T-cell lymphoma [24,29]; and chronic inflammatory diseases, such as vitiligo, alopecia, and psoriasis [30,31].

In this review, we provide an overview of the general characteristics of T_RM_. Then, narrowing our focus to skin T_RM_ in humans, we summarize the involvement of skin T_RM_ in cutaneous disorders, especially psoriasis. We also mention the possibility of engaging T_RM_ as a disease index and treatment target in psoriasis. Since CD8^+^ T_RM_ are the best-characterized population, we focus on CD8^+^ T_RM_ and describe this population as T_RM_ in this review unless otherwise mentioned.

## 2. The Characteristics of T_RM_

T cells in the neonatal murine skin are predominant with dendritic epidermal T cells (DETCS) with restricted antigenic specificity [32], and neonatal human skin holds only a few T cells [24]. Thus, T_RM_ are assumed to develop from circulating T cells according to repeated antigen exposure. In the local inflammation caused by specific antigens, the robustly expanded effector T cells emerge in the circulation and the affected tissues, and both T_CM_ and T_RM_ are assumed to arise from a part of these effector T cells [25,33].

The general characteristics of T_RM_ across the tissues include the loss of migration and the gain of retention. The development and maintenance of these characteristics in T_RM_ are driven by complex factors, such as cytokine and chemokine receptors, the other cell-surface molecules being responsible for tissue homing and retention, and transcription factors (Figure 1).

### 2.1. Cell Surface Molecules

While homing molecules including chemokine receptors are diverse depending on the target peripheral tissues, the molecules related to tissue retention seem to be shared among various tissues. In general, T_RM_ lack the expression of the secondary lymphoid homing molecules CC-chemokine receptor 7 (CCR7) and L-selectin, which are expressed on T_CM_ and naïve T cells [1]. The tissue retention molecules CD69 and CD103 (αE integrin) are widely recognized as the markers for T_RM_. CD103 is a ligand of E-cadherin that is expressed on epithelial cells [34], and CD69 interferes with sphingosine-1-phosphate (S1P) receptor-1, which allows the cells to exit from peripheral tissues by sensing the density of S1P [35]. CD69 also reportedly regulates the uptake of L-tryptophan and the intracellular quantity of L-tryptophan-derived activator of the aryl hydrocarbon receptor (AhR) [36], which is reportedly involved in the persistence of T_RM_ [32]. These functions would explain at least partially the importance of these molecules in tissue retention. However, their expression varies, possibly depending on the tissues and the causes of T_RM_ development. T_RM_ lacking CD103 expression have been described in some peripheral tissues and secondary lymphoid organs [37,38] and CD103^+^ T_RM_ can be found in the dermis and adult central nervous system where E-cadherin is absent, implying that binding to E-cadherin is not required for the persistence of T_RM_ in peripheral tissues [24,39]. Although CD69 is expressed on the majority of T_RM_ in various peripheral tissues, T_RM_ negative for CD69 expression are also noted [33]. We thus have to take into account that these two molecules are not able to cover T_RM_ universally.

### 2.2. Transcription Factors

Transcriptional regulation is also presumably common among T_RM_ in various tissues. For instance, the expression of AhR is increased in skin T_RM_ as compared with naïve T cells and splenic T cells, possibly favoring the maintenance of skin T_RM_ [32]. Rapamycin inhibits the formation of T_RM_ in the intestinal and vaginal mucosa, highlighting a positive link of mammalian target of rapamycin and the downstream transcription factors with the formation of T_RM_ [40]. The maintenance of lung T_RM_ may be related to Notch signaling, including the upregulation of the downstream transcription factor RBPJ [41]. The augmented uptake of exogenous lipids accompanied by the upregulation of fatty acid binding proteins (FABPs) 4 and 5 is one of the characteristic processes involved in the generation and maintenance of skin T_RM_ [42]. Hypoxia-inducible factor-1α, which is a transcription factor in the downstream of FABP5 signaling, reportedly promotes the residency and anti-tumor function of tumor-infiltrating T cells in the murine malignancy model [43]. The downregulation of T-box transcription factors T-bet and EOMES [44] and the upregulation of Blimp-1, Hobit [45], and Runx3 [46,47] have also been reported to be involved in the differentiation and/or maintenance of T_RM_.

### 2.3. Skin-Homing Molecules

In addition to the shared characteristics of various T_RM_, skin T_RM_ are shown to have their own homing molecules. As one of skin’s homing molecules, cutaneous lymphocyte-associated antigen (CLA) binds to E-selectin and P-selectin and allows the cells to migrate into skin [23]. The chemokine receptors CCR4, CCR8, CCR10, CXCR3, and CXCR6 are also regarded as important skin-homing and/or retention molecules for at least some skin T cells [16,48,49,50,51,52].

### 2.4. Fate Decision of T_RM_

How the fate of T_RM_ differentiation is decided remains an unsolved question. T_RM_ reportedly derive from circulating T cells lacking high expression of the killer cell lectin-like receptor subfamily G member 1 (KLRG1), which is regarded as a terminal differentiation marker [16,47]. Another report demonstrates that the effector T cells with enriched expression of T_RM_-associated genes, such as *Itgae* (CD103), *Itga1* (CD49a), *Cd101*, *Ahr*, and *Fabp5*, already exist as memory precursor cells and preferentially differentiate into T_RM_ [53], suggesting that the fate of T_RM_ is at least partially decided in the early stage of adoptive immune memory formation. On the other hand, the time-course single-cell RNA-sequencing analysis in a murine model with lymphocytic-choriomeningitis-virus infection revealed that the transcriptional characteristics of T_RM_ can be detected from gut-infiltrating T cells at the earliest 4 days after infection, and the characteristics are distinct from those found in splenic T cells [54], implying that the T_RM_ differentiation program is initiated after the cells enter the specific peripheral tissues. Further elucidation of the T_RM_ differentiation mechanism will require further research.

## 3. Human Skin T_RM_

In general, human T_RM_ and murine T_RM_ share core transcriptional, phenotypic, and functional profiles, including the almost global expression of CD69 and dominant CD103 expression in CD8 fractions [45,55,56,57]. In patients with cutaneous T-cell lymphoma (CTCL), the treatment with alemtuzumab, which depletes circulating T cells and spares the T_RM_, does not result in serious infection [58], implying the role of skin T_RM_ in protective immunity. The T_RM_ phenotype of the malignant cells in CTCL is related to the clinical manifestation of well-demarcated lesions, suggesting that the sessile property of T_RM_ also exists in humans [24]. In vitro experiments suggest skin T_RM_ maintain the production of IL-17A and IFN-γ in reaction with pathogen challenges through aging [59]. Using transcriptomic and functional data, human T_RM_ are found to abolish their senescent phenotype and survive for over 10 years in specific circumstances [46], replicating the longevity of T_RM_ in humans.

However, T_RM_ in humans are presumably more diverse and widely distributed. For instance, CD4^+^ T_RM_ are found in both the epidermis and dermis in humans, although murine skin CD4^+^ T_RM_ are predominantly found in dermis [17,24,60,61]. T_RM_ are also found in secondary lymphoid organs, such as the spleen, lymph nodes, and tonsils in humans [55,56].

The factors that may cause the difference between human skin T_RM_ properties and those observed in laboratory mice may include the following: (1) the thick epidermis with abundant niche for T_RM_ [24,62]; (2) the low density of hair follicles that express cytokines important for T_RM_ migration and survival, including IL-7 and IL-15 [63,64]; (3) the frequent exposure to foreign antigens; (4) the small population of γδT cells with the lack of DETC in the human epidermis [65] (however, we do not know whether the recently identified αβγδT cell population in fetal skin can replace DETC) [66]. The longer survival period of human T_RM_ compared to murine life span [46] may also cause difficulty in adapting the findings in murine models to human biology.

The involvement of skin T_RM_ is highlighted in chronic inflammatory disorders and cutaneous malignancies. In the lesional skin of alopecia areata, T_RM_ with the ability to produce granzyme B are dominant and related to disease prognosis, implying their involvement in the pathogenesis [67]. Intraepidermal IFN-γ-producing T_RM_ are enriched in the cured sites of fixed drug eruption [26], suggesting the contribution of this fraction to the reproducible property. In patients with atopic dermatitis, cutaneous T_RM_ with the production of IL-4 and IL-13 are also indicated to be involved in the disease pathogenesis [68]. Dermal T_RM_ are increased with the production potential of perforin, granzyme B, and IFN-γ in vitiligo [30,69], which are presumably specific for melanocyte antigens. In malignant melanoma, skin T_RM_ provide protection against tumor regrowth and are involved in vitiligo formation, suggestive of their specific reactivity against melanoma antigens [70]. Better understanding of cutaneous T_RM_ will pave the way for novel management and treatment of skin diseases.

The methodologies for evaluating skin T_RM_ are summarized in Table 1. In the translational research field, one of the most popular methods for analyzing T_RM_ is fluorescence-activated cell sorting (FACS) analysis. However, conducting this method from biopsied skin specimens is not practical in the daily clinical settings considering the burden for both patients and clinicians. Immunohistochemistry (IHC) and/or immunofluorescence (IF) for T_RM_-related molecules, such as CD3, CD8, CD69, and CD103, on the residual biopsy specimens carried out for diagnosis is probably more feasible to date. To establish non-invasive methods for predicting the activities of skin T_RM_, such as analyzing tape-stripped or surface-swabbed samples, will require further research.

## 4. Skin T_RM_ in the Pathogenesis of Psoriasis

Psoriasis, hereafter referred to as plaque psoriasis, is an immune-mediated chronic inflammatory skin disorder characterized by well-demarcated persistent scaly indurated erythematous plaques. The contributions of environment [71], hereditary predisposition [72], and autoantigens [73] are implied to be involved in disease development. Circulating T cells were previously regarded as responsible for the lesion formation in psoriasis. However, the inhibition of E-selectin, which is required for T-cell migration from the blood stream to skin, was noted to be ineffective [74]. Another blocking strategy of T-cell migration by the biologics targeting CD11a also did not show dramatic efficacy [75]. However, in a humanized murine model where psoriatic nonlesional skin specimens are grafted to immunodeficient mice [76], the healthy-appearing nonlesional skin grafts spontaneously develop psoriatic disease, suggesting that the cells residing in the nonlesional skin are sufficient for the development of psoriatic disease. These results have led to the theory that T_RM_ may play a crucial role in the pathogenesis of psoriasis.

The fate of skin T_RM_ is affected by the skin microenvironment, and in psoriasis, this is also the case. Several skin-constituting factors have been reported to support the development and persistence of IL-17A-producing T_RM_ in psoriasis. Keratinocytes in disease-naïve sites of psoriasis upregulate the expression of chemokines, such as CCL20 upon stimulation by skin commensal fungi [77]. Since CCL20 is a ligand for CCR6, which is a signature molecule of IL-17A-producing T cells, the activated keratinocytes in the disease-naïve sites of psoriasis are to recruit IL-17A-producing T cells to the disease-naïve sites, leading to the accumulation of IL-17A-producing T_RM_ [77]. In turn, IL-17A from T_RM_ stimulates keratinocytes to express CCL20, further accelerating the recruitment of CCR6^+^ cells [78]. In the resolved skin, the continuous production of IL-23 and IL-15 from Langerhans cells presumably support the maintenance of IL-17A-producing T_RM_ in the epidermis [79]. The reduced repertoire of IL-17A-producing T cells in the resolved skin, which has been observed in different psoriatic patients, implies the existence of common antigens that drive the accumulation of psoriatic T_RM_ [80]. Several potential autoantigens have been reported in psoriasis (Figure 2). For example, cationic antimicrobial peptide LL-37 produced by various cells including keratinocytes binds self-DNA and triggers the activation of plasmacytoid dendritic cells (pDC) and TNF/iNOS-producing dendritic cells (TIP-DC) [81,82]. A disintegrin-like and metalloprotease domain containing thrombospondin type 1 motif-like 5 (ADAMTSL5) in complex with HLA-C*06:02 on the surface of melanocytes confers epidermal CD8^+^ T-cell response [83]. Neo-lipid antigens generated by phospholipase A2 group 4D (PLA2G4D) from mast cells and keratinocytes trigger the CD1a-reactive T cells to produce IL-17A and IL-22 [84]. Keratin 17, a human epidermal keratin that shares a sequential homology with streptococcal M protein, is recognized by HLA-Cw*0602-restricted IFN-γ-producing CD8^+^ T cells [85,86]. Taken together, these results suggest the synchronizing roles of the skin microenvironment in the development and persistence of pathogenic cutaneous T_RM_.

In the lesional skin of patients with psoriasis, T_RM_ consist of both CD4 and CD8 fractions, which synchronize the elevated immune response by the increased expression of inflammatory cytokines, such as IL-17A, IL-22, and IFN-γ [62,80,87,88]. While IL-17A-producing CD4^+^ T_RM_ also exist in healthy skin, the enrichment of CD8^+^ T_RM_ producing IL-17A in the epidermis is one of the characteristics of psoriasis [87,88]. In disease-naïve skin that has never experienced disease formation, IL-17A production is augmented by T_RM_ [77], and the increase in IL-17A-producing CD8^+^ T_RM_ at the dispense of IFN-γ-producing T_RM_ occurs according to disease duration [88].

IFN-γ-producing T_RM_ are also dominant in the epidermis and express the complex of CD49a–CD29, also known as very late antigen 1 (VLA-1) or α1β1 integrin [76]. CD49a^+^ T_RM_ are involved in the pathogenesis of psoriasis. The number of epidermal CD8^+^CD49a^+^ T_RM_ correlates with the severity of the disease [89], and an experimental blockade of CD49a in mice transplanted with psoriatic skin reduces the disease formation [76]. However, since the blockade of whole CD8^+^ T cells almost completely prevents disease development in the similar psoriatic skin-engrafted murine model [90], CD49a^+^ T_RM_ with IFN-γ production are not likely the key population for disease development, while the CD8^+^ T cell population likely includes a critical fraction for disease pathogenesis. In fact, CD8^+^ T_RM_ without the expression of CD49a are defined as an IL-17A-producing T_RM_ subset [30].

Successful treatment with an IL-17A-targeting biologics results in a decreased number of IL-17A-producing T_RM_ in resolved skin, but the frequency of these cells is not altered within the remaining T cells [91]. Another study on residual psoriasis after the use of biologics revealed a decrease in keratinocyte proliferation. However, the percentage of IL-17A-producing CD103^+^ T_RM_ was not significantly reduced after the treatments [92]. Similarly, a new normal in the persistence of IL-17A-producing T_RM_ with CCR6 and IL-23R expression in the resolved skin has been established [62,80]. IL-17A-producing CD8^+^ T_RM_ and IL-22-producing CD4^+^ T_RM_ remain in the psoriatic epidermis for as long as six years after starting the successful TNF-α-targeting treatment [62]. Taken together, these findings highlight the essential standing point of IL-17A-producing T_RM_ as one of the pathogenic populations of skin T_RM_ in psoriasis.

## 5. Targeting Skin T_RM_ in the Management of Psoriasis

Regardless of the persistence of this population by various treatments in psoriasis, many of the current and upcoming therapeutics in clinical practice presumably exert an indirect influence on cutaneous IL-17A-producing T_RM_. Since the remission period after successful treatments inversely correlates with the relative IL-17 signaling of the resolved skin compared to IL-10 and IFN-γ signaling [93], the relative reduction, if not elimination, of IL-17A-producing T_RM_ may be of help in controlling psoriatic disease activity.

Biologics targeting the IL-17 pathway reportedly reduce IL-17 signaling and the amount of T cells in the lesion [94]. Furthermore, the biologics targeting IL-23 decrease this fraction from the lesion more strongly compared to those targeting IL-17A [95]. Ultraviolet irradiation leads to the diminishment of IL-17A-producing T cells in skin [96], and this T-cell fraction includes T_RM_. Topical vitamin D analogues and corticosteroids reportedly reduce the lesional IL-17A-producing T_RM_, possibly including pathogenic T_RM_ [97,98]. Retinoic acid prevents Th17 differentiation and possibly promotes the properties of regulatory T cells [99,100]. As the oral phosphodiesterase 4 inhibitor (PDE4i) diminishes the pro-inflammatory cytokine production from circulating T cells [101], the function of both topical and systemic PDE4i could be revisited from the perspective of skin T_RM_. An AhR agonist modulates the Th17 property of T cells, and the efficacy of its topical form possibly affects IL-17A-producing T cells in skin, including T_RM_ [102].

Proof-of-concept approaches that directly and exclusively target pathogenic populations of T_RM_ should be subjected to further studies. The candidate strategies might include the inhibition of the pathways involved in IL-15 signaling to perturb the survival of pathogenic T_RM_ and the blockade of the pathways processing fatty acids to suppress the lipid metabolism of pathogenic T_RM_. Targeting the transcription factors specified for differentiation and maintenance of pathogenic T_RM_ is also an attractive strategy. However, although the risk of targeting these populations of T_RM_ is unknown, it may cause the loss of local immune memory against pathogens in the skin. Since the characteristic cell surface molecules and transcription factors found in T_RM_ properties can be overlapped with the sessile properties of other cell types, such as innate lymphoid cells and B cells [103,104], the strategies targeting T_RM_ might also affect the other tissue-sessile immunity. Specific treatment targets for psoriatic dysfunctional T_RM_, excluding the other T_RM_ and skin-resident immune cells, would be ideal.

## 6. Conclusions

Extensive studies with rigorous methodologies have broadened our knowledge on T_RM_ in general and those residing in the skin in particular (Table 1). The involvement of skin T_RM_ in the pathogenesis of skin diseases is also being elucidated. Several key points are highlighted below:T_RM_ originate from circulating T cells, do not recirculate, and provide the first line of adaptive cellular defense in the residing tissues.The functional skew of skin T_RM_ is indicated in chronic skin inflammatory diseases.In psoriasis, IL-17-A-producing CD8^+^ T_RM_ may be among the pathogenic populations in the skin.Pathogenic populations of skin T_RM_ can be targeted in the current and future treatments of psoriasis. Skin T_RM_ can also serve as a potential index of the disease.

Further studies on T_RM_ will advance the management of not only psoriasis but other diseases in which this subset of T cells plays a role. jcm-10-03822-t001_Table 1Table 1Several major findings related to methodologies used in research on humans.Key FindingsMajor Methodologies
A role of skin T_RM_ in protective immunity in humansFACS[58]Skin T_RM_ with the potential of producing cytokines are infiltrated in the lesion of patients with GVHDFC, single-cell TCR sequencing, and IF[46]Cells residing in nonlesional skin are sufficient, and the recruitment of circulating cells is not necessary for the development of psoriatic diseaseTransplantation, FC, quantitative RT-PCR, and IHC[76]CD8^+^ T_RM_ producing IL-17A in the epidermis is one of the characteristics in psoriasisFC and IHC[87]The increase in IL-17A-producing CD8^+^ T_RM_ during the distribution of IFN-γ-producing T_RM_ occurs according to psoriasis durationFC and IF[88]The successful treatment with IL-17A-targeting biologics results in a decreased number of IL-17A-producing CD8^+^ T_RM_ in resolved psoriatic skin, but the frequency of these cells is not alteredFC, IHC, and IF[91]IL-17A-producing CD8^+^ T_RM_ and IL-22-producing CD4^+^ T_RM_ remain in the psoriatic epidermis for as long as six years after starting the successful TNF-α-targeting treatmentFC, quantitative RT-PCR, and IF[62]FC: flow cytometry, TCR: T-cell receptor, RT-PCR: reverse transcription polymerase chain reaction, IF: immunofluorescence, IHC: immunohistochemistry.

## Figures and Tables

**Figure 1 jcm-10-03822-f001:**
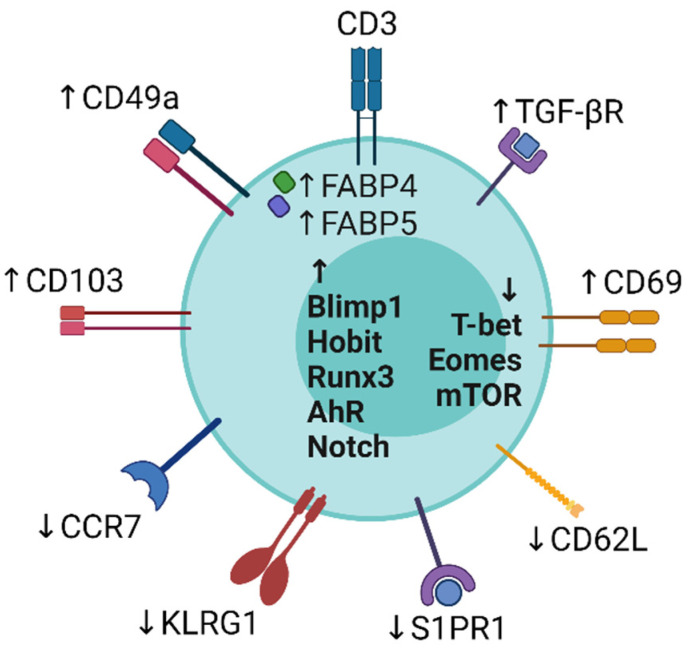
A. Surface markers, intracellular molecules, and transcription factors of T_RM_. The expression levels of these molecules on T_RM_ are shown by upward arrows (increased expressions) and downward arrows (decreased expressions). Created with BioRender.com (accessed on 21 August 2021).

**Figure 2 jcm-10-03822-f002:**
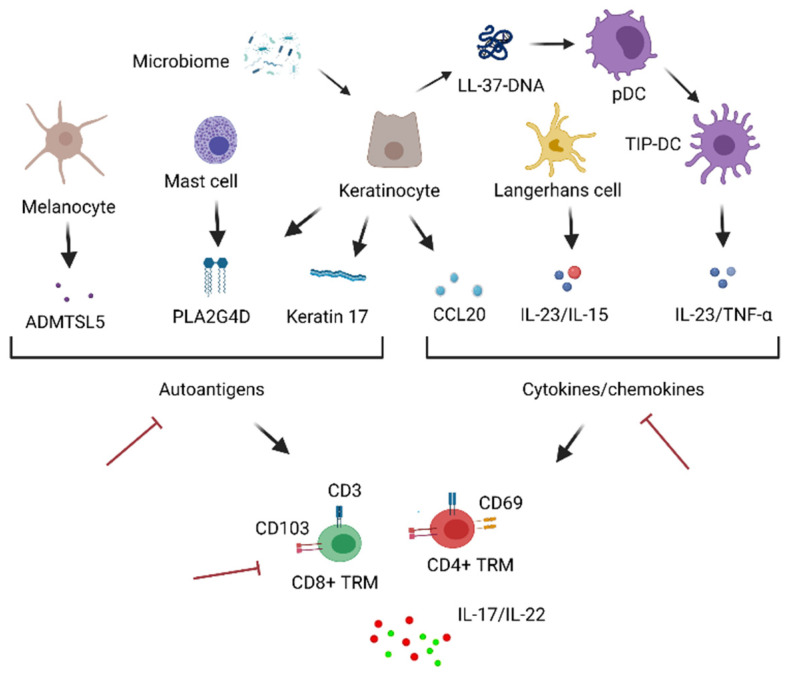
Development of T_RM_ in psoriasis. T_RM_ are activated by either autoantigens or cytokines/chemokines. Autoantigens include ADMTSL5 on the surface of melanocytes, PLAG4D from mast cells and keratinocytes, and keratin 17 from keratinocytes. Antimicrobial peptide LL-37, also from keratinocytes, binds to self-DNA to activate pDC and TIP-DC, leading to the production of IL-23/TNF-α. IL-23/15 from Langerhans cells and CCL20 from keratinocytes also activate T_RM_. These stimulated T_RM_ produce proinflammatory cytokines, such as IL-17A and IL-22, the hallmarks of psoriasis. The development of pathogenic T_RM_ can be inhibited by stopping pathways related to T_RM_ activation or directly inhibiting the activity of T_RM_ (red inhibition icon). Created with BioRender.com (assessed on 21 August 2021).

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
