# Peer review of "Skin-Resident Memory T Cells: Pathogenesis and Implication for the Treatment of Psoriasis"

_jcm, 2021, doi:10.3390/jcm10173822_

Round 1
Reviewer 1 Report
The topic selected by Authors for review seems to be current, interesting and intriguing and is a good choice with interest for JCM readers.
In my opinion a good review should explain basic concepts for someone who is new in the field. The readers of this special issue will be clinicians treating patients diagnosed with psoriasis. After reading this rewiev they should understand the idea, and arguments that favor new target as a therapeutic option. I think that at this stage autors missed this goal.
I would suggest at least
- To provide proper introduction including historical concept, lines 79-86 in my opinion would be more suitable for introduction
- How the subset might be characterised in clinical paractice
- Broader analysis of physiologicala function of these population in humans
- What situation and circumstances afects residual memory T cells in physiology - it is just mention
- More focused discussion on current evidence for blocking T residual memory cells
- There is no any information of possible disadvantages or risk with this strategy
- In conclusion more information of future needs and directions
- Table which summarised current results with methodology used in the reaserch would be useful
- Figure - the indentification of resident memory T should be given
Author Response
Reviewer #1
The topic selected by Authors for review seems to be current, interesting and intriguing and is a good choice with interest for JCM readers.
In my opinion a good review should explain basic concepts for someone who is new in the field. The readers of this special issue will be clinicians treating patients diagnosed with psoriasis. After reading this rewiev they should understand the idea, and arguments that favor new target as a therapeutic option. I think that at this stage autors missed this goal.
- We largely appreciate all your comments and apologize for the generally inadequate summarization not considering the readers’ point of views. We tried to improve the manuscript according to your instructive suggestions as below.
I would suggest at least
- To provide proper introduction including historical concept, lines 79-86 in my opinion would be more suitable for introduction
- We totally agree with the comment. We reorganized the manuscript in 6 sections: introduction, the characteristics of TRM, human skin TRM, skin TRM in the pathogenesis of psoriasis, targeting skin TRM in the management of psoriasis, and conclusion. In the section 1 “Introduction”, we reviewed how the concept of resident memory T cells are established and moved a part of the sentences in the previous lines 79-86 to this new section for more straightforward understanding (lines 49-51). The rest of the sentences are expanded in the new section 3 “Human skin TRM”.
- How the subset might be characterised in clinical practice
- We appreciate this practical question which grabs attention of clinicians in the field. We agree that one of the most popular methods to characterize this subset would be flow cytometry of skin cells, which is very difficult to conduct in the daily clinical settings. We would suggest the immunohistochemistry and or immunofluorescence analysis for TRM-related markers such as CD3, CD8, CD69, and CD103 using the remaining biopsy specimens when we need the biopsy for the diagnosis. We also added the description that we need to establish the non-invasive methods for predicting the activity of this fraction by finding the TRM-associated molecules which can be detected from the tape-stripped or swabbed samples. These descriptions are added in lines 183-191 in section 3 “Human skin TRM”.
- Broader analysis of physiologicala function of these population in humans
- We appreciate this instructive suggestion. In humans, the researches for elucidating the functional activities of TRM start from finding the differences of TRM in disease conditions including infectious diseases, immune-related disorders, and cutaneous malignancies. We added the summarization of the findings in these conditions in lines 169-181 of the section 3 “Human skin TRM”.
- What situation and circumstances afects residual memory T cells in physiology - it is just mention
- We apologize for the insufficient description. We admit that the information related to the situations and circumstances affecting the development of TRM is scattered and scarce, and reorganized the manuscript so that the information is collected in the new section 2 “The characteristics of TRM” (lines 77-138). We grouped the influencing factors into cell surface molecules, transcription factors, and skin-homing molecules, and added the explanation on each reported molecules with new citations.
- More focused discussion on current evidence for blocking T residual memory cells
- We appreciate your instructive advice. We rearranged the paragraphs and sentences in the new section 4 “Skin TRM in the pathogenesis of psoriasis” and 5 “Targeting skin TRM in the management of psoriasis” focus on the current evidence on the relation of existing treatments with the blockade of pathogenic TRM In section 4, we overviewed the background of regarding TRM population as a pathogenic population in psoriasis, followed by an emphasis on skin microenvironment that affect the formation and persistence of the pathogenic TRM. Then we summarized the findings on how this population is affected or unaffected by various treatments in section 5. We hope this order will be helpful for the clinicians not familiar with the topic.
- There is no any information of possible disadvantages or risk with this strategy
- Thank you very much for pointing out, and we apologize for missing this critical information. The largest concerning for targeting TRM in the treatment is probably the loss of immune memories in skin especially against pathogens. Thus, the treatment strategy ideally should be specific for the pathogenic TRM. We added the description on this point in the revised section 5“Targeting skin TRM in the management of psoriasis”, lines 310-317.
- In conclusion more information of future needs and directions
- Thank you very much for the advice. We made a new section 5 “Targeting skin TRM in the management of psoriasis”, elaborating current therapeutics that may exert influence on skin TRM in psoriasis. We then propose approaches that directly target pathogenic populations of TRM. It should be specific to the pathways involved in the development of dysfunctional fractions of TRM. As with the disease index, we need to establish repeatable non-invasive methods for sampling to predict the activity of TRM (in lines 189-191).
- Table which summarised current results with methodology used in the reaserch would be useful
- We appreciate the comments and totally agree that the summarization of the methodology for the evaluation of TRM would be of use. We inserted Table 1 summarizing the findings and methodologies used in the researches on human psoriasis.
- Figure - the indentification of resident memory T should be given
- Thank you very much for your instructive suggestion. We changed figure 1 to the illustration of the expressed surface molecules and transcription factors in TRM based on the information on the section 2.

Reviewer 2 Report
The Review presented by Vu et al provides a broad overview of tissue-resident memory T cells and explores their contribution to the pathology of psoriasis. While I believe this is an interesting area of discussion, I have concerns regarding the execution and content of the Review that would need to be addressed prior to reconsidering this manuscript for publication.
- On multiple occasions, the authors state phrases that are not supported by literature or can be misleading to the reader. For example, Line 27 “Trm are believed to destroy target cells through perforin/granzyme apoptosis” – no in vivo study has demonstrated this point. Line 36-38, “The augmented uptake of exogenous lipid is one of the characteristic processes involved in the generation and maintenance of Trm” – this has only been demonstrated for skin Trm. Lines 46-47 “…mice show CD69+CD103- Trm are not present in small intestine, which is in contrast to those in skin where CD69+CD103- Trm are detected” – this is not always the case, such as in HSV infection. Line 71 “The density of hair follicles, crucial for residency of Trm, is lower” – one would argue that it’s simply a correlation and not a prerequisite as indicated by the word ‘crucial’. Line 101 “The slow proliferation may also support the longevity of Trm” – this phrase appears is not substantiated by further explanations or references. While I understand the authors intent, more care needs to be taken in the description of published work to ensure readers outside of the field are not misled.
- References are inaccurate and not representative of the literature. Throughout the text, the references do not correctly cite original work that led to the discussed results. While referencing other reviews is beneficial for well accepted information, specific studies that demonstrate a key finding need to be directly acknowledged. For example, inaccurate references are observed in line 42 (ref 13), line 47 (ref 14), line 51 (refs 15 and 16) and others.
- Structure of the Review could be improved to ensure all readers can follow the topic. For example, there was no introduction to psoriasis as a disease and its current treatment options. For examples, after the authors discuss skin Trm, it might be beneficial to give a broad summary of the role of these cells in multiple disorders, before honing in on psoriasis and its current biology. The ‘implications of Trm’ could also be expanded upon as they were only summarised in one short paragraph (lines 197-203) at the end of the Review.
- Clarification of CD4 or CD8 Trm throughout the text would greatly benefit the reader's understanding of the topic
- Figure 2 was difficult to follow and didn’t accurately represent the text. The figure could benefit from being divided into several panels for easier digestion, but also, expanding on the concepts of targeting Trm using a pictorial approach.
Author Response
Reviewer #2
The Review presented by Vu et al provides a broad overview of tissue-resident memory T cells and explores their contribution to the pathology of psoriasis. While I believe this is an interesting area of discussion, I have concerns regarding the execution and content of the Review that would need to be addressed prior to reconsidering this manuscript for publication.
- We appreciate your critical review of our manuscript and sincerely apologize for our inadequate unscientific summarization of this topic.
- On multiple occasions, the authors state phrases that are not supported by literature or can be misleading to the reader.
- We sincerely apologize for our unscientific/misleading description and insufficient references. We learnt from these errors and revised according to your detailed suggestion as follows. We also reorganized the manuscript with 6 sections in order for the smoother understanding: introduction, the characteristics of TRM, human skin TRM, skin TRM in the pathogenesis of psoriasis, targeting skin TRM in the management of psoriasis, and conclusion.
- For example, Line 27 “Trm are believed to destroy target cells through perforin/granzyme apoptosis” – no in vivo study has demonstrated this point.
- We apologize for the incorrect overstatement. Thanks to your comments, we realized that we expanded many in vitro observations to the assumption of in vivo We decided to make a general introduction and erased the vague information based on in vitro observation including these pointed sentences.
- Line 36-38, “The augmented uptake of exogenous lipid is one of the characteristic processes involved in the generation and maintenance of Trm” – this has only been demonstrated for skin Trm.
- Thank you very much for pointing out this misleading information, and we apologize for this incorrect description. We indicate the restriction of the sentence to skin TRM. We moved the sentences to the new section 2 “The characteristics of TRM” (lines 106-109).
- Lines 46-47 “…mice show CD69+CD103- Trm are not present in small intestine, which is in contrast to those in skin where CD69+CD103- Trm are detected” – this is not always the case, such as in HSV infection.
- We sincerely apologize for these misleading explanations. Thanks to your comments, we realized that covering all the cases analyzed in various experimental conditions would not help the readers to gain the general understanding of this topic. We decided to limit the description of the difference between mice and humans to general description as in lines 91-94 of the new section 2 “The characteristics of TRM”.
- Line 71 “The density of hair follicles, crucial for residency of Trm, is lower” – one would argue that it’s simply a correlation and not a prerequisite as indicated by the word ‘crucial’.
- We appreciate this suggestion, too. We also realize that your suggestion should be applied to the whole manuscript. We tried to avoid the unscientific explanations which cause the misleading of causal association from the whole manuscript, including this previous line 71. This line is now in lines 161-162 of the revised document.
- Line 101 “The slow proliferation may also support the longevity of Trm” – this phrase appears is not substantiated by further explanations or references. While I understand the authors intent, more care needs to be taken in the description of published work to ensure readers outside of the field are not misled.
- Thank you very much for your instructive comment, and we apologize again for our insufficient misleading description. We decided to omit the assumption lacking enough background, including this expression.
- References are inaccurate and not representative of the literature. Throughout the text, the references do not correctly cite original work that led to the discussed results. While referencing other reviews is beneficial for well accepted information, specific studies that demonstrate a key finding need to be directly acknowledged. For example, inaccurate references are observed in line 42 (ref 13), line 47 (ref 14), line 51 (refs 15 and 16) and others.
- We sincerely apologize for this scarce manuscript and largely appreciate your patient instruction. We omitted the description related to line 42 (ref 13) and line 47 (ref 14). Line 51 (ref 15 16) are deleted and substituted with original articles according to cell surface molecules and transcription factors indicated (lines 77-114 of the revised section 2). We also find other inaccurate references in the previous lines 30-31, line 92, and line 108. We put references relating to the original articles and add information accordingly.
- Structure of the Review could be improved to ensure all readers can follow the topic. For example, there was no introduction to psoriasis as a disease and its current treatment options. For examples, after the authors discuss skin Trm, it might be beneficial to give a broad summary of the role of these cells in multiple disorders, before honing in on psoriasis and its current biology. The ‘implications of Trm’ could also be expanded upon as they were only summarised in one short paragraph (lines 197-203) at the end of the Review.
- We appreciate the comments of the reviewer about the structure. We reorganized the manuscript with 6 sections including introduction. A brief introduction of psoriasis is inserted into the beginning of section 4 “Skin TRM in the pathogenesis of psoriasis” (lines 210-213). A broad summary of the role of TRM in multiple disorders is given in section 3 “human skin TRM”. We also add information on TRM in fixed drug eruption, along with vitiligo and atopic dermatitis (169-181). We added a new section 5 “Targeting skin TRM in the management of psoriasis” and summarized the effect of current/trialed treatments on TRM, and included small suggestion in the implication of TRM as a treatment target.
- Clarification of CD4 or CD8 Trm throughout the text would greatly benefit the reader's understanding of the topic
- We appreciate your comments and we totally agree that clarification of CD4+ and CD8+ TRM must be done critically, We apologize again for our scarce explanation. Since the most findings on TRM dysfunction in cutaneous disorders are restricted to CD8 TRM, we decided to focus on CD8+ TRM in our revised manuscript. We added the explanation in the first section line 60-62 that we focus on CD8 fraction, and we noted the description on CD4 when needed.
- Figure 2 was difficult to follow and didn’t accurately represent the text. The figure could benefit from being divided into several panels for easier digestion, but also, expanding on the concepts of targeting Trm using a pictorial approach.
- We totally admit your point. Thank you very much for the suggestion. We delete the skin background so that more focus can be made on TRM We also separate cytokines/chemokines from autoantigens and rearrange the related cells and the molecules they produce. Captions that provide explanations to the figure are also provided.

Reviewer 3 Report
The review manuscript by Vu, Koguchi-Yoshioka, and Watanabe is a well-written piece that successfully juggles insightful description of recent developments and summarises current knowledge in the field of resident memory T cells in psoriasis. The authors start by providing a brief overview of TRMs and skin TRMs, and then waste no time in delving into their role in the pathogenesis of psoriasis. They describe how their role in disease was proven, what it is they do, how they are recruited to the skin, how they are maintained in the skin, and what potential therapeutic strategies could be employed.
A few brief points that the authors may want to consider:
- The manuscript could benefit from an outline of the key points addressed in the abstract.
- The strategies for targeting TRMs could be further expanded upon (last paragraph of section 3). For instance, are any of them being trialled for other indications? Is there any evidence that they are implicated in psoriasis (from successful therapeutics)?
A few minor points:
- A few typos have been highlighted in the text
- Line 15, and Line 87: the authors talk about a specific circumstance. If this is not a typo, I would state that specific circumstance, otherwise I would change to “specific circumstances”.
- Line 34: could the authors explain the sentence by either rephrasing it, or adding further detail?
- Line 47: I may be mistaken, but I believe that reference [14] doesn’t go into whether CD103 constitutes a TRM marker of fidelity to the intestine over the skin. Perhaps could the authors double-check whether this was the paper they meant to refer to?
- Lines 187 to 193: while the grammar is correct, this sentence may benefit from being broken down in several parts. Furthermore, seen as these autoantigens are (mostly) expressed in situ, at the site of psoriatic lesions, this provides a basis for the recurrence and persistence of psoriasis. Do the authors believe that this review could benefit from the expansion of this section?

Author Response
Reviewer #3
The review manuscript by Vu, Koguchi-Yoshioka, and Watanabe is a well-written piece that successfully juggles insightful description of recent developments and summarises current knowledge in the field of resident memory T cells in psoriasis. The authors start by providing a RM pathogenesis of psoriasis. They describe how their role in disease was proven, what it is they do, how they are recruited to the skin, how they are maintained in the skin, and what potential therapeutic strategies could be employed.
- We largely appreciate your positive comments to our manuscript.
A few brief points that the authors may want to consider:
- The manuscript could benefit from an outline of the key points addressed in the abstract.
- Thank you very much for your instructive suggestion. We totally agree that the outline of the key points will be beneficial to readers. The abstract and conclusion are now revised to indicate the key points addressed in the manuscript.
- The strategies for targeting TRMs could be further expanded upon (last paragraph of section 3). For instance, are any of them being trialled for other indications? Is there any evidence that they are implicated in psoriasis (from successful therapeutics)?
- We appreciate your suggestive comments. We expand the last paragraph of the previous section 3 to the independent section 5 “Targeting skin TRM in the management of psoriasis”. We add information on the current/trialed therapeutics such as vitamin D3, retinoic acid, PDE4 inhibitors, and AhR agonist in addition to the currently used biologics (lines 284-317).
A few minor points:
- A few typos have been highlighted in the text
- We are sorry for the careless typos. We read the text again for any typos and made corrections.
- Line 15, and Line 87: the authors talk about aspecific circumstance. If this is not a typo, I would state that specific circumstance, otherwise I would change to “specific circumstances”.
- Thank you for pointing out this English-grammar error. We opt for “specific circumstances” and make changes accordingly.
- Line 34: could the authors explain the sentence by either rephrasing it, or adding further detail?
- We apologize for the unclear sentence. Thanks to your suggestions, we realized that this vague information does not help understand the process of cell migration into skin. We rephrased and reorganized the sentence into the context of relation between TRM and circulating populations of T cells. It is now in lines 37-40 of the new section 1 (introduction).
- Line 47: I may be mistaken, but I believe that reference [14] doesn’t go into whether CD103 constitutes a TRMmarker of fidelity to the intestine over the skin. Perhaps could the authors double-check whether this was the paper they meant to refer to?
- We apologize again for this careless incorrectness. Thanks to the suggestion, we realized that pointing out each difference depending on various analysis would just cause confusion to the readers. We omit the expression and modify the manuscript so that we can clarify the general characteristics of TRM (in the new section 2) and then those of skin TRM (in the new section 3).
- Lines 187 to 193: while the grammar is correct, this sentence may benefit from being broken down in several parts. Furthermore, seen as these autoantigens are (mostly) expressed in situ, at the site of psoriatic lesions, this provides a basis for the recurrence and persistence of psoriasis. Do the authors believe that this review could benefit from the expansion of this section?
- We appreciate your comment. As you kindly suggest, these autoantigens activate not only TRM but also other cell type in the lesion skin such as dendritic cells and Langerhans cells. We give more details in the lines 238-247 of the revised section 4. It now reads as following:
For example, cationic antimicrobial peptide LL-37 produced by various cells including keratinocytes binds self-DNA and trigger the activation of plasmacytoid dendritic cells. A disintegrin-like and metalloprotease domain containing thrombospondin type 1 motif-like 5 (ADAMTSL5) in complex with HLA-C*06:02 on the surface of melanocytes confers epidermal CD8+ T-cell response. Neolipid antigens generated by phospholipase A2 group 4D (PLA2G4D) from mast cells and keratinocytes trigger specific CD1a-reactive T cells inducing production of IL-17A and IL-22. Keratin 17, a human epidermal keratin that share a sequence homology with streptococcal M protein, is recognized by HLA-Cw*0602-restricted IFN-γ-producing CD8+ T cells.

Round 2
Reviewer 1 Report
In my opinion the corrected version of the paper is significantly improved.
The only remark - I did not find any reference to Fig 1. in the text.
Author Response
In my opinion the corrected version of the paper is significantly improved.
- Thank you very much again for sparing your time and giving us instructive suggestions.
The only remark - I did not find any reference to Fig 1. in the text.
- We are sorry about lacking the note. We added the reference of (Figure 1) in the line 75 of the section 2 “The characteristics of TRM”.

Reviewer 2 Report
The revised manuscript by Vu et al has sufficiently addressed the concerns and now presents with a well structured and clear summary on this important topic.
The only two minor points to make is i) the term 'tissue' in 'tissue TRM' is not necessary as TRM have already been defined as 'tissue-resident memory T cells' and ii) I would remove 'Ki67' from Figure 1 as highly proliferative skin Trm has only been shown in the context of antigen recall and not as their 'phenotype' per se.
Aside from these points, I commend the authors for significantly improving the manuscript that I feel is now suitable for publication.
Author Response
The revised manuscript by Vu et al has sufficiently addressed the concerns and now presents with a well structured and clear summary on this important topic.
- We appreciate all your instructive comments and suggestion again.
The only two minor points to make is i) the term 'tissue' in 'tissue TRM' is not necessary as TRM have already been defined as 'tissue-resident memory T cells' and ii) I would remove 'Ki67' from Figure 1 as highly proliferative skin Trm has only been shown in the context of antigen recall and not as their 'phenotype' per se.
- Thank you so much for your careful suggestion, and we are sorry, again, for our unscientific description. i) We absolutely agree with your suggestion, and omitted “tissue” from the abbreviation “tissue TRM” in lines 57, 117, and 145. ii) We also totally agree with your suggestion. We meant to include Ki67 as one of the downregulated transcription factors. However, as you point out, the expression of Ki67 should be depending on the cell activation status and the various expression levels should be detected from each type of memory T cells. The term Ki67 is now removed from Figure 1. Also, we apologize that FABP4 and FABP5 were incorrectly included in the transcription factors in the previous Figure 1. We fixed this issue in this revision, too.
Aside from these points, I commend the authors for significantly improving the manuscript that I feel is now suitable for publication.
- Thank you very much again for your time and thoughtful comments.